# Flexible statistical inference for mechanistic models of neural dynamics

**Jan-Matthis Lueckmann**[*1], **Pedro J. Gonçalves**[*1], **Giacomo Bassetto**[1],
**Kaan Öcal**[1,2], **Marcel Nonnenmacher**[1], **Jakob H. Macke**[†1]
[1] research center caesar, an associate of the Max Planck Society, Bonn, Germany
[2] Mathematical Institute, University of Bonn, Bonn, Germany
{jan-matthis.lueckmann, pedro.goncalves, giacomo.bassetto,
kaan.oecal, marcel.nonnenmacher, jakob.macke}@caesar.de

## Abstract

Mechanistic models of single-neuron dynamics have been extensively studied in computational neuroscience. However, identifying which models can quantitatively reproduce empirically measured data has been challenging. We propose to overcome this limitation by using likelihood-free inference approaches (also known as Approximate Bayesian Computation, ABC) to perform full Bayesian inference on single-neuron models. Our approach builds on recent advances in ABC by learning a neural network which maps features of the observed data to the posterior distribution over parameters. We learn a Bayesian mixture-density network approximating the posterior over multiple rounds of adaptively chosen simulations. Furthermore, we propose an efficient approach for handling missing features and parameter settings for which the simulator fails, as well as a strategy for automatically learning relevant features using recurrent neural networks. On synthetic data, our approach efficiently estimates posterior distributions and recovers ground-truth parameters. On in-vitro recordings of membrane voltages, we recover multivariate posteriors over biophysical parameters, which yield model-predicted voltage traces that accurately match empirical data. Our approach will enable neuroscientists to perform Bayesian inference on complex neuron models without having to design model-specific algorithms, closing the gap between mechanistic and statistical approaches to single-neuron modelling.

## 1 Introduction

Biophysical models of neuronal dynamics are of central importance for understanding the mechanisms by which neural circuits process information and control behaviour. However, identifying which models of neural dynamics can (or cannot) reproduce electrophysiological or imaging measurements of neural activity has been a major challenge [1]. In particular, many models of interest – such as multi-compartment biophysical models [2], networks of spiking neurons [3] or detailed simulations of brain activity [4] – have intractable or computationally expensive likelihoods, and statistical inference has only been possible in selected cases and using model-specific algorithms [5, 6, 7]. Many models are defined implicitly through *simulators*, i.e. a set of dynamical equations and possibly a description of sources of stochasticity [1]. In addition, it is often of interest to identify models which can reproduce particular *features* in the data, e.g. a firing rate or response latency, rather than the full temporal structure of a neural recording.

---

[*]Equal contribution

[†]Current primary affiliation: Centre for Cognitive Science, Technical University Darmstadt

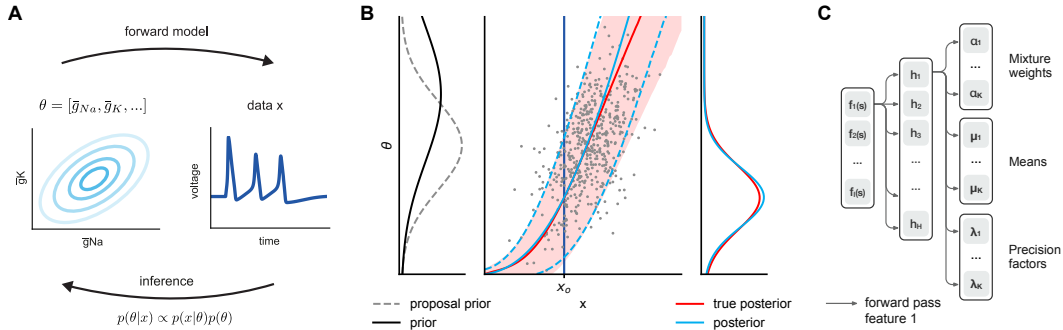

Figure 1: **Flexible likelihood-free inference for models of neural dynamics. A.** We want to flexibly and efficiently infer the posterior over model parameters given observed data, on a wide range of models of neural dynamics. **B.** Our method approximates the true posterior on $\theta$ around the observed data $\mathbf{x}_o$ by performing density estimation on data simulated using a proposal prior. **C.** We train a Bayesian mixture-density network (MDN) for posterior density estimation.

In the absence of likelihoods, the standard approach in neuroscience has been to use heuristic parameter-fitting methods [2, 8, 9]: distance measures are defined on multiple features of interest, and brute-force search [10, 11] or evolutionary algorithms [2, 9, 12, 13] (neither of which scales to high-dimensional parameter spaces) are used to minimise the distances between observed and model-derived features. As it is difficult to trade off distances between different features, the state-of-the-art methods optimise multiple objectives and leave the final choice of a model to the user [2, 9]. As this approach is not based on statistical inference, it does not provide estimates of the full posterior distribution – thus, while this approach has been of great importance for identifying 'best fitting' parameters, it does not allow one to identify the full space of parameters that are consistent with data and prior knowledge, or to incrementally refine and reject models.

Bayesian inference for likelihood-free simulator models, also known as Approximate Bayesian Computation [14, 15, 16], provides an attractive framework for overcoming these limitations: like parameter-fitting approaches in neuroscience [2, 8, 9], it is based on comparing summary features between simulated and empirical data. However, unlike them, it provides a principled framework for full Bayesian inference and can be used to determine how to trade off goodness-of-fit across summary statistics. However, to the best of our knowledge, this potential has not been realised yet, and ABC approaches are not used for linking mechanistic models of neural dynamics with experimental data (for an exception, see [17]). Here, we propose to use ABC methods for statistical inference of mechanistic models of single neurons. We argue that ABC approaches based on conditional density estimation [18, 19] are particularly suited for neuroscience applications.

We present a novel method (Sequential Neural Posterior Estimation, SNPE) in which we sequentially train a mixture-density network across multiple rounds of adaptively chosen simulations[1]. Our approach is directly inspired by prior work [18, 19], but overcomes critical limitations: first, a flexible mixture-density network trained with an importance-weighted loss function enables us to use complex proposal distributions and approximate complex posteriors. Second, we represent a full posterior over network parameters of the density estimator (i.e. a "posterior on posterior-parameters") which allows us to take uncertainty into account when adjusting weights. This enables us to perform 'continual learning', i.e. to effectively utilise all simulations without explicitly having to store them. Third, we introduce an approach for efficiently dealing with simulations that return missing values, or which break altogether – a common situation in neuroscience and many other applications of simulator-based models – by learning a model that predicts which parameters are likely to lead to breaking simulations, and using this knowledge to modify the proposal distribution. We demonstrate the practical effectiveness and importance of these innovations on biophysical models of single neurons, on simulated and neurophysiological data. Finally, we show how recurrent neural networks can be used to directly learn relevant features from time-series data.

## 1.1 Related work using likelihood-free inference for simulator models

Given experimental data $\mathbf{x}_o$ (e.g. intracellular voltage measurements of a single neuron, or extracellular recordings from a neural population), a model $p(\mathbf{x}|\boldsymbol{\theta})$ parameterised by $\boldsymbol{\theta}$ (e.g. biophysical parameters, or connectivity strengths in a network simulation) and a prior distribution $p(\boldsymbol{\theta})$, our goal is to perform statistical inference, i.e. to find the posterior distribution $\hat{p}(\boldsymbol{\theta}|\mathbf{x} = \mathbf{x}_o)$. We assume that the model $p(\mathbf{x}|\boldsymbol{\theta})$ is only defined through a *simulator* [14, 15]: we can generate samples $\mathbf{x}_n \sim \mathbf{x}|\boldsymbol{\theta}$ from it, but not evaluate $p(\mathbf{x}|\boldsymbol{\theta})$ (or its gradients) explicitly. In neural modelling, many models are defined through specification of a dynamical system with external or intrinsic noise sources or even through a black-box simulator (e.g. using the NEURON software [20]).

In addition, and in line with parameter-fitting approaches in neuroscience and most ABC techniques [14, 15, 21], we are often interested in capturing summary statistics of the experimental data (e.g. firing rate, spike-latency, resting potential of a neuron). Therefore, we can think of $\mathbf{x}$ as resulting from applying a feature function $f$ to the raw simulator output $\mathbf{s}$, $\mathbf{x} = f(\mathbf{s})$, with $\dim(\mathbf{x}) \ll \dim(\mathbf{s})$.

Classical ABC algorithms simulate from multiple parameters, and reject parameter sets which yield data that are not within a specified distance from the empirically observed features. In their basic form, proposals are drawn from the prior ('rejection-ABC' [22]). More efficient variants make use of a Markov-Chain Monte-Carlo [23, 24] or Sequential Monte-Carlo (SMC) samplers [25, 26]. Sampling-based ABC approaches require the design of a distance metric on summary features, as well as a rejection criterion ($\varepsilon$), and are exact only in the limit of small $\varepsilon$ (i.e. many rejections) [27], implying strong trade-offs between accuracy and scalability. In SMC-ABC, importance sampling is used to sequentially sample from more accurate posteriors while $\varepsilon$ is gradually decreased.

Synthetic-likelihood methods [28, 21, 29] approximate the likelihood $p(\mathbf{x}|\boldsymbol{\theta})$ using multivariate Gaussians fitted to repeated simulations given $\boldsymbol{\theta}$ (see [30, 31] for generalisations). While the Gaussianity assumption is often motivated by the central limit theorem, distributions over features can in practice be complex and highly non-Gaussian [32]. For example, neural simulations sometimes result in systematically missing features (e.g. spike latency is undefined if there are no spikes), or diverging firing rates.

Finally, methods originating from regression correction [33, 18, 19] simulate multiple data $\mathbf{x}_n$ from different $\boldsymbol{\theta}_n$ sampled from a proposal distribution $\tilde{p}(\boldsymbol{\theta})$, and construct a conditional density estimate $q(\boldsymbol{\theta}|\mathbf{x})$ by performing a regression from simulated data $\mathbf{x}_n$ to $\boldsymbol{\theta}_n$. Evaluating this density model at the observed data $\mathbf{x}_o$, $q(\boldsymbol{\theta}|\mathbf{x}_o)$ yields an estimate of the posterior distribution. These approaches do not require parametric assumptions on likelihoods or the choice of a distance function and a tolerance ($\varepsilon$) on features. Two approaches are used for correcting the mismatch between prior and proposal distributions: Blum and François [18] proposed the importance weights $p(\boldsymbol{\theta})/\tilde{p}(\boldsymbol{\theta})$, but restricted themselves to proposals which were truncated priors (i.e. all importance weights were 0 or 1), and did not sequentially optimise proposals over multiple rounds. Papamakarios and Murray [19] recently used stochastic variational inference to optimise the parameters of a mixture-density network, and a post-hoc division step to correct for the effect of the proposal distribution. While highly effective in some cases, this closed-form correction step can be numerically unstable and is restricted to Gaussian and uniform proposals, limiting both the robustness and flexibility of this approach. SNPE builds on these approaches, but overcomes their limitations by introducing four innovations: a highly flexible proposal distribution parameterised as a mixture-density network, a Bayesian approach for continual learning from multiple rounds of simulations, and a classifier for predicting which parameters will result in aborted simulations or missing features. Fourth, we show how this approach, when applied to time-series data of single-neuron activity, can automatically learn summary features from data.

## 2 Methods

### 2.1 Sequential Neural Posterior Estimation for likelihood-free inference

In SNPE, our goal is to learn the parameters $\phi$ of a posterior model $q_\phi(\boldsymbol{\theta}|\mathbf{x} = f(\mathbf{s}))$ which, when evaluated at $\mathbf{x}_o$, approximates the true posterior $p(\boldsymbol{\theta}|\mathbf{x}_o) \approx q_\phi(\boldsymbol{\theta}|\mathbf{x} = \mathbf{x}_o)$. Given a prior $p(\boldsymbol{\theta})$, a proposal prior $\tilde{p}(\boldsymbol{\theta})$, pairs of samples $(\boldsymbol{\theta}_n, \mathbf{x}_n)$ generated from the proposal prior and the simulator, and a calibration kernel $K_\tau$, the posterior model can be trained by minimising the importance-weighted log-loss

$$\mathcal{L}(\phi) = -\frac{1}{N} \sum_n \frac{p(\boldsymbol{\theta}_n)}{\tilde{p}(\boldsymbol{\theta}_n)} K_\tau(\mathbf{x}_n, \mathbf{x}_o) \log q_\phi(\boldsymbol{\theta}_n|\mathbf{x}_n), \tag{1}$$

as is shown by extending the argument in [19] with importance-weights $p(\boldsymbol{\theta}_n)/\tilde{p}(\boldsymbol{\theta}_n)$ and a kernel $K_\tau$ in Appendix A.

Sampling from a proposal prior can be much more effective than sampling from the prior. By including the importance weights in the loss, the analytical correction step of [19] (i.e. division by the proposal prior) becomes unnecessary: SNPE directly estimates the posterior density rather than a conditional density that is reweighted post-hoc. The analytical step of [19] has the advantage of side-stepping the additional variance brought about by importance-weights, but has the disadvantages of (1) being restricted to Gaussian proposals, and (2) the division being unstable if the proposal prior has higher precision than the estimated conditional density.

The calibration kernel $K_\tau(\mathbf{x}, \mathbf{x}_o)$ can be used to calibrate the loss function by focusing it on simulated data points $\mathbf{x}$ which are close to $\mathbf{x}_o$ [18]. Calibration kernels $K_\tau(\mathbf{x}, \mathbf{x}_o)$ are to be chosen such that $K_\tau(\mathbf{x}_o, \mathbf{x}_o) = 1$ and that $K_\tau$ decreases with increasing distance $\|\mathbf{x} - \mathbf{x}_o\|$, given a bandwidth $\tau^2$. Here, we only used calibration kernels to exclude bad simulations by assigning them kernel value zero. An additional use of calibration kernels would be to limit the accuracy of the posterior density estimation to a region near $\mathbf{x}_o$. Choice of the bandwidth implies a bias-variance trade-off [18]. For the problems we consider here, we assumed our posterior model $q_\phi(\boldsymbol{\theta}|\mathbf{x})$ based on a multi-layer neural network to be sufficiently flexible, such that limiting bandwidth was not necessary.

We sequentially optimise the density estimator $q_\phi(\boldsymbol{\theta}|\mathbf{x}) = \sum_k \alpha_k \mathcal{N}(\boldsymbol{\theta}|\boldsymbol{\mu}_k, \boldsymbol{\Sigma}_k)$ by training a mixture-density network (MDN) [19] with parameters $\phi$ over multiple 'rounds' $r$ with adaptively chosen proposal priors $\tilde{p}^{(r)}(\boldsymbol{\theta})$ (see Fig. 1). We initialise the proposal prior at the prior, $\tilde{p}^{(1)}(\boldsymbol{\theta}) = p(\boldsymbol{\theta})$, and subsequently take the posterior of the previous round as the next proposal prior (Appendix B). Our approach is not limited to Gaussian proposals, and in particular can utilise multi-modal and heavy-tailed proposal distributions.

## 2.2 Training the posterior model with stochastic variational inference

To make efficient use of simulation time, we want the posterior network $q_\phi(\boldsymbol{\theta}|\mathbf{x})$ to use *all* simulations, including ones from previous rounds. For computational and memory efficiency, it is desirable to avoid having to store all old samples, or having to train a new model at each round. To achieve this goal, we perform Bayesian inference on the weights $\mathbf{w}$ of the MDN across rounds. We approximate the distribution over weights as independent Gaussians [34, 35]. Note that the parameters $\phi$ of this Bayesian MDN are are means and standard deviations per each weight, i.e., $\phi = \{\phi_m, \phi_s\}$. As an extension to the approach of [19], rather than assuming a zero-centred prior over weights, we use the posterior over weights of the previous round, $\pi_{\phi^{(r-1)}}(\mathbf{w})$, as a prior for the next round. Using stochastic variational inference, in each round, we optimise the modified loss

$$
\begin{aligned}
\mathcal{L}(\boldsymbol{\phi}^{(r)}) = &-\frac{1}{N}\sum_n \frac{p(\boldsymbol{\theta}_n)}{\tilde{p}^{(r)}(\boldsymbol{\theta}_n)} K_\tau(\mathbf{x}_n, \mathbf{x}_o) \big\langle \log q_{\mathbf{w}}(\boldsymbol{\theta}_n|\mathbf{x}_n) \big\rangle_{\pi_{\boldsymbol{\phi}^{(r)}}(\mathbf{w})} \\
&+ \frac{1}{N} D_{\mathrm{KL}}\left(\pi_{\boldsymbol{\phi}^{(r)}}(\mathbf{w}) || \pi_{\boldsymbol{\phi}^{(r-1)}}(\mathbf{w})\right).
\end{aligned}
\tag{2}
$$

Here, the distributions $\pi(\mathbf{w})$ are approximated by multivariate normals with diagonal covariance. The continuity penalty ensures that MDN parameters that are already well constrained by previous rounds are less likely to be updated than parameters with large uncertainty (see Appendix C). In practice, gradients of the expectation over networks are approximated using the local reparameterisation trick [36].

## 2.3 Dealing with bad simulations and bad features, and learning features from time series

**Bad simulations:** Simulator-based models, and single-neuron models in particular, frequently generate nonsensical data (which we name 'bad simulations'), especially in early rounds in which the relevant region of parameter space has not yet been found. For example, models of neural dynamics can easily run into self-excitation loops with diverging firing rates [37] (Fig. 4A). We introduce a feature $b(\mathbf{s}) = 1$ to indicate that $\mathbf{s}$ and $\mathbf{x}$ correspond to a bad simulation. We set $K(\mathbf{x}_n, \mathbf{x}_o) = 0$

whenever $b(\mathbf{x}_n) = 1$ since the density estimator should not spend resources on approximating the posterior for bad data. With this choice of calibration kernel, bad simulations are ignored when updating the posterior model – however, this results in inefficient use of simulations.

We propose to learn a model $\hat{g} : \boldsymbol{\theta} \to [0, 1]$ to predict the probability that a simulation from $\boldsymbol{\theta}$ will break. While any probabilistic classifier could be used, we train a binary-output neural network with log-loss on $(\boldsymbol{\theta}_n, b(\mathbf{s}_n))$. For each proposed $\boldsymbol{\theta}$, we reject $\boldsymbol{\theta}$ with probability $\hat{g}(\boldsymbol{\theta})$, and do not carry out the expensive simulation[3]. The rejections could be incorporated into the importance weights (which would require estimating the corresponding partition function, or assuming it to be constant across rounds), but as these rejections do not depend on the data $\mathbf{x}_o$, we interpret them as modifying the prior: from an initially specified prior $p(\boldsymbol{\theta})$, we obtain a modified prior excluding those parameters which likely will lead to nonsensical simulations. Therefore, the predictive model $\hat{g}(\boldsymbol{\theta})$ does not only lead to more efficient inference (especially in strongly under-constrained scenarios), but is also useful in identifying an effective prior – the space of parameters deemed plausible a priori intersected with the space of parameters for which the simulator is well-behaved.

**Bad features:** It is frequently observed that individual features of interest for fitting single-neuron models cannot be evaluated: for example, the spike latency cannot be evaluated if a simulation does not generate spikes, but the fact that this feature is missing might provide valuable information (Fig. 4C). SNPE can be extended to handle 'bad features' by using a carefully designed posterior network. For each feature $f_i(\mathbf{s})$, we introduce a binary feature $m_i(\mathbf{s})$ which indicates whether $f_i$ is missing. We parameterise the input layer of the posterior network with multiplicative terms of the form $h_i(\mathbf{s}) = f_i(\mathbf{s}) \cdot (1 - m_i(\mathbf{s})) + c_i \cdot m_i(\mathbf{s})$ where the term $c_i$ is to be learned. This approach effectively learns an imputation value $c_i$ for each missing feature. For a more expressive model, one could also include terms which learn interactions across different missing-feature indicators and/or features, but we did not explore this here.

**Learning features:** Finally, we point out that using a neural network for posterior estimation yields a straightforward way of *learning* relevant features from data [38, 39, 40]. Rather than feeding summary features $f(\mathbf{s})$ into the network, we directly feed time-series recordings of neural activity into the network. The first layer of the MDN becomes a recurrent layer instead of a fully-connected one. By minimising the variational objective (Eq.2), the network learns informative summary features about posterior densities.

## 3   Results

While SNPE is in principle applicable to any simulator-based model, we designed it for performing inference on models of neural dynamics. In our applications, we concentrate on single-neuron models. We demonstrate the ability of SNPE to recover ground-truth posteriors in Gaussian Mixtures and Generalised Linear Models (GLMs) [41], and apply SNPE to a Hodgkin-Huxley neuron model and an autapse model, which can have parameter regimes of unstable behaviour and missing features.

### 3.1   Statistical inference on simple models

**Gaussian mixtures:** We first demonstrate the effectiveness of SNPE for inferring the posterior of mixtures of two Gaussians, for which we can analytically compute true posteriors. We are interested in the numerical stability of the method ('robustness') and the 'flexibility' to approximate multi-modal posteriors. To illustrate the robustness of SNPE, we apply SNPE and the method proposed by [19] (which we refer to by Conditional Density Estimation for Likelihood-free Inference, CDE-LFI) to infer the common mean of a mixture of two Gaussians, given samples from the mixture distribution (Fig. 2A; details in Appendix D.1). Whereas SNPE works robustly across multiple algorithmic rounds, CDE-LFI can become unstable: its analytical correction requires a division by a Gaussian which becomes unstable if the precision of the Gaussian does not increase monotonically across rounds (see 2.1). Constraining the precision-matrix to be non-decreasing fixes the numerical issue, but leads to biased estimates of the posterior. Second, we apply both SNPE and CDE-LFI to infer the two means of a mixture of two Gaussians, given samples $\mathbf{x}$ from the mixture distribution (Fig. 2B; Appendix D.1). While SNPE can use bi-modal proposals, CDE-LFI cannot, implying reduced efficiency of proposals on strongly non-Gaussian or multi-modal problems.

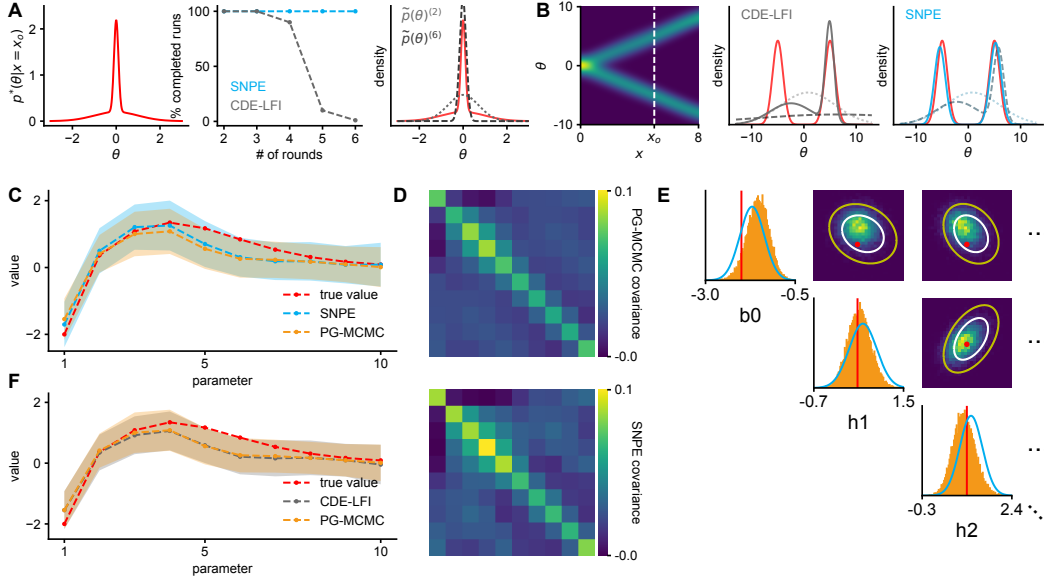

Figure 2: **Inference on simple statistical models. A.** Robustness of posterior inference on 1-D Gaussian Mixtures (GMs). Left: true posterior given observation at $\mathbf{x}_o = 0$. Middle: percentage of completed runs as a function of number of rounds; SNPE is robust. Right: Gaussian proposal priors tend to underestimate tails of posterior (red). **B.** Flexibility of posterior inference. Left: True posterior for 1-D bimodal GM and observation $\mathbf{x}_o$. Middle and right: First round proposal priors (dotted), second round proposal priors (dashed) and estimated posteriors (solid) for CDE-LFI and SNPE respectively (true posterior red). SNPE allows multi-modal proposals. **C, F.** Application to GLM. Posterior means and variances are recovered well by both CDE-LFI and SNPE. For reference, we approximate the posterior using likelihood-based PG-MCMC. **D.** Covariance matrices for SNPE and PG-MCMC. **E.** Partial view of the posterior for 3 out of 10 parameters (all 10 parameters in Appendix G). Ground-truth parameters in red. 2-D marginals for SNPE (lines) and PG-MCMC (histograms). White and yellow contour lines correspond to 68% and 95% of the mass, respectively.

**Generalised linear models:** Generalised linear models (GLM) are commonly used to model neural responses to sensory stimuli. For these models, several techniques are available to estimate the posterior distribution over parameters, making them ideally suited to test SNPE in a single-neuron model. We evaluated the posterior distribution over the parameters of a GLM using a Pólya-Gamma sampler (PG-MCMC, [42, 43]) and compared it to the posterior distributions estimated by SNPE (Appendix D.2 for details). We found a good agreement of the posterior means and variances (Fig. 2C), covariances (Fig. 2D), as well as pairwise marginals (Fig. 2E). We note that, since GLMs have close-to-Gaussian posteriors, the CDE-LFI method works extremely well on this problem (Fig. 2F).

In summary, SNPE leads to accurate and robust estimation of the posterior in simple models. It works effectively even on multi-modal posteriors on which CDE-LFI exhibits worse performance. On a GLM-example with an (almost) Gaussian posterior, the CDE-LFI method works extremely well, but SNPE yields very similar posterior estimates (see Appendix F for additional comparison with SMC-ABC).

### 3.2 Statistical inference on Hodgkin-Huxley neuron models

**Simulated data:** The Hodgkin-Huxley equations [44] describe the dynamics of a neuron's membrane potential and ion channels given biophysical parameters (e.g. concentration of sodium and potassium channels) and an injected input current (Fig. 3A, see Appendix D.3). We applied SNPE to a Hodgkin-Huxley model with channel kinetics as in [45] and inferred the posterior over 12 biophysical parameters, given 20 voltage features of the simulated data. The true parameter values are close to the mode of the inferred posterior (Fig. 3B, D), and in a region of high posterior probability. Samples from the posterior lead to voltage traces that are similar to the original data, supporting the correctness of the approach (Fig. 3C).

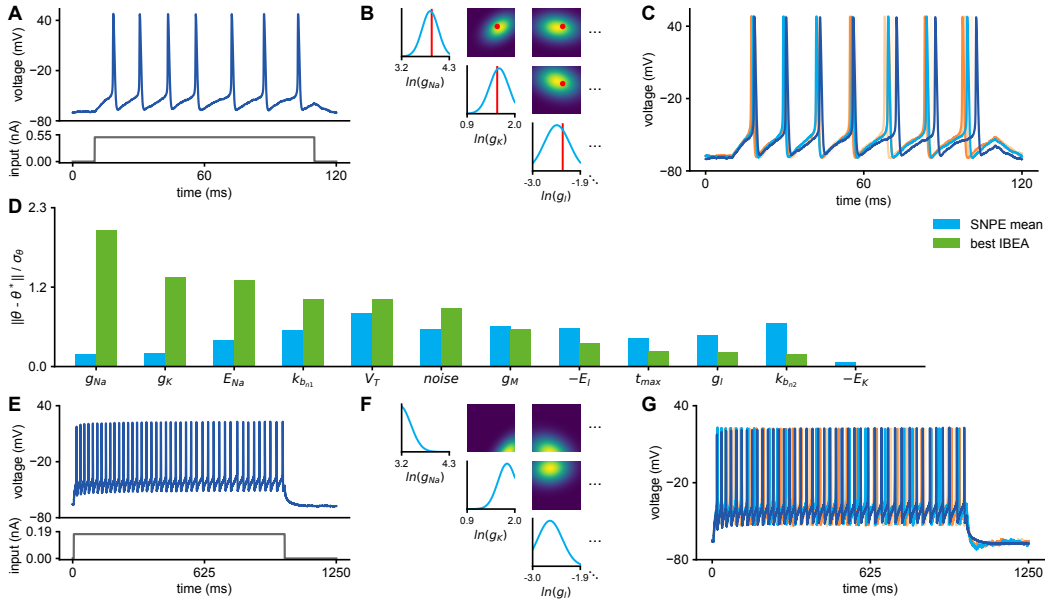

Figure 3: **Application to Hodgkin-Huxley model: A.** Simulation of Hodgkin-Huxley model with current injection. **B.** Posterior over 3 out of 12 parameters inferred with SNPE (12 parameters in Appendix G). True parameters have high posterior probabilities (red). **C.** Traces for the mode (cyan) of and samples (orange) from the inferred posterior match the original data (blue). **D.** Comparison between SNPE and a standard parameter-fitting procedure based on a genetic algorithm, IBEA: difference between the mode of SNPE or IBEA best parameter set, and the ground-truth parameters, normalised by the standard deviations obtained by SNPE. **E-G.** Application to real data from Allen Cell Type Database. Inference over 12 parameters for cell *464212183*. Results presented as in A-C.

Biophysical neuron models are typically fit to data with genetic algorithms applied to the distance between simulated and measured data-features [2, 8, 9, 46]. We compared the performance of SNPE with a commonly used genetic algorithm (Indicator Based Evolutionary Algorithm, IBEA, from the BluePyOpt package [9]), given the same number of model simulations (Fig. 3D). SNPE is comparable to IBEA in approximating the ground-truth parameters – note that defining an objective measure to compare the two approaches is difficult, as they both minimise different criteria. However, unlike IBEA, SNPE also returns a full posterior distribution, i.e. the space of all parameters consistent with the data, rather than just a 'best fit'.

**In-vitro recordings:** We also applied the approach to *in vitro* recordings from the mouse visual cortex (see Appendix D.4, Fig. 3E-G). The posterior mode over 12 parameters of a Hodgkin-Huxley model leads to a voltage trace which is similar to the data, and the posterior distribution shows the space of parameters for which the output of the model is preserved. These posteriors could be used to motivate further experiments for constraining parameters, or to study invariances in the model.

### 3.3 Dealing with bad simulations and features

**Bad simulations:** We demonstrate our approach (see Section 2.3) for dealing with 'bad simulations' (e.g. for which firing rates diverge) using a simple, two-parameter 'autapse' model for which the region of stability is known. During SNPE, we concurrently train a classifier to predict 'bad simulations' and update the prior accordingly. This approach does not only lead to a more efficient use of simulations, but also identifies the parameter space for which the simulator is well-defined, information that could be used for further model analysis (Fig. 4A, B).

**Bad features:** Many features of interest in neural models, e.g. the latency to first spike after the injection of a current input, are only well defined in the presence of other features, e.g. the presence of spikes (Fig. 4C). Given that large parts of the parameter space can lead to non-spiking behaviour, missing features occur frequently and cannot simply be ignored. We enriched our MDN with an extra layer which imputes values to the absent features, values which are optimised alongside the rest of the parameters of the network (Fig. 4D; Appendix E). Such imputation has marginal computational

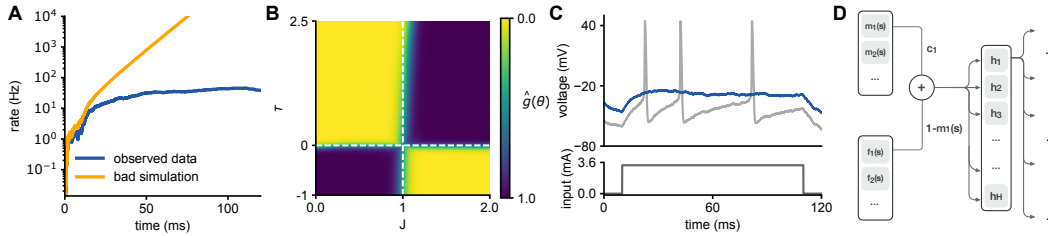

Figure 4: **Inference on neural dynamics has to deal with diverging simulations and missing features. A.** Firing rate of a model neuron connected to itself (autapse). If the strength of the self-connection (parameter $J$) is bigger than 1, the dynamics are unstable (orange line - bad simulation). **B.** Portion of parameter space leading to diverging simulations learned by the classifier (yellow: low probability of bad simulation, blue: high probability), and comparison with analytically computed boundaries (white, see Appendix D.5). **C.** Illustration of a model neuron in two parameter regimes, spiking (grey trace) and non-spiking (blue). When the neuron does not spike, features that depend on the presence of spiking, such as the latency to first spike, are not defined. **D.** Our MDN is augmented with a multiplicative layer which imputes values for missing features.

cost and grants us the convenience of not having to hand-tune imputation values, or to reject all simulations for which any individual feature might be missing.

**Learning features with recurrent neural networks (RNNs):** In neural modelling, it is often of interest to work with hand-designed features that are thought to be particularly important or informative for particular analysis questions [2]. For instance, the shape of the action potential is intimately related to the dynamics of sodium and potassium channels in the Hodgkin-Huxley model. However, the space of possible features is immense, and given the highly non-linear nature of many of the neural models in question, it can sometimes be of interest to simply perform statistical inference without having to hand-design features. Our approach provides a straightforward means of doing that: we augment the MDN with a RNN which runs along the recorded voltage trace (and stimulus, here a coloured-noise input) to learn appropriate features to constrain the model parameters. As illustrated in figure 5B, the first layer of the network, which previously received pre-computed summary statistics as inputs, is replaced by a recurrent layer that receives full voltage and current traces as inputs. In order to capture long-term dependencies in the sequence input, we use gated-recurrent units (GRUs) for the RNN [47]. Since we are using 25 GRU units and only keep the final output of the unrolled RNN (many-to-one), we introduce a bottleneck. The RNN thus transforms the voltage trace and stimulus into a set of 25 features, which allow SNPE to recover the posterior over the 12 parameters (Fig. 5C). As expected, the presence of spikes in the observed data leads to a tighter posterior for parameters associated to the main ion channels involved in spike generation, $E_{Na}$, $E_K$, $g_{Na}$ and $g_K$.

## 4  Discussion

Quantitatively linking models of neural dynamics to data is a central problem in computational neuroscience. We showed that likelihood-free inference is at least as general and efficient as 'black-box' parameter fitting approaches in neuroscience, but provides full statistical inference, suggesting it to be the method of choice for inference on single-neuron models. We argued that ABC approaches based on density estimation are particularly useful for neuroscience, and introduced a novel algorithm (SNPE) for estimating posterior distributions. We can flexibly and robustly estimate posterior distributions, even when large regions of the parameter space correspond to unstable model behaviour, or when features of choice are missing. Furthermore, we have extended our approach with RNNs to automatically define features, thus increasing the potential for better capturing salient aspects of the data with highly non-linear models. SNPE is therefore equipped to estimate posterior distributions under common constraints in neural models.

Our approach directly builds on a recent approach for density estimation ABC (CDE-LFI, [19]). While we found CDE-LFI to work well on problems with unimodal, close-to-Gaussian posteriors and stable simulators, our approach extends the range of possible applications, and these extensions are critical for the application to neuron models. A key component of SNPE is the proposal prior, which guides the sampling on each round of the algorithm. Here, we used the posterior on the previous round as the proposal for the next one, as in CDE-LFI and in many Sequential-MC approaches. Our

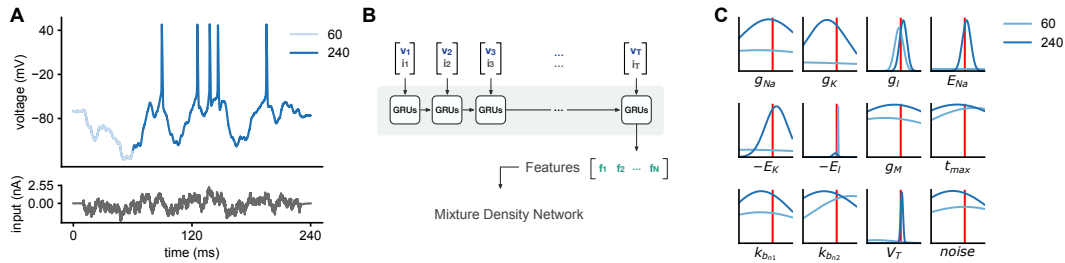

Figure 5: **We can learn informative features using a recurrent mixture-density network (R-MDN). A.** We consider a neuron driven by a colored-noise input current. **B.** Rather than engineering summary features to reduce the dimensionality of observations, we provide the complete voltage trace and input current as input to an R-MDN. The unrolled forward pass is illustrated, where a many-to-one recurrent network reduces the dimensionality of the inputs ($T$ time steps long) to a feature vector of dimensionality $N$. **C.** Our goal is to infer the posterior density for two different observations: (1) the full 240ms trace shown in panel A; and (2) the initial 60ms of its duration, which do not show any spike. We show the obtained marginal posterior densities for the two observations, using a 25-dimensional feature vector learned by the RNN. In the presence of spikes, the posterior uncertainty gets tighter around the true parameters related to spiking.

method could be extended by alternative approaches to designing proposal priors [48, 49], e.g. by exploiting the fact that we also represent a posterior over MDN parameters: for example, one could design proposals that guide sampling towards regions of the parameter space where the uncertainty about the parameters of the posterior model is highest. We note that, while here we concentrated on models of single neurons, ABC methods and our approach will also be applicable to models of populations of neurons. Our approach will enable neuroscientists to perform Bayesian inference on complex neuron models without having to design model-specific algorithms, closing the gap between mechanistic and statistical models, and enabling theory-driven data-analysis [50].

## Acknowledgements

We thank Maneesh Sahani, David Greenberg and Balaji Lakshminarayanan for useful comments on the manuscript. This work was supported by SFB 1089 (University of Bonn) and SFB 1233 (University of Tübingen) of the German Research Foundation (DFG) to JHM and by the caesar foundation.

## Footnotes

[1]Code available at https://github.com/mackelab/delfi

[2]While we did not investigate this here, an attractive idea would be to base the kernel of the distance between $\mathbf{x}_n$ and $\mathbf{x}_o$ on the divergence between the associated posteriors, e.g. $K_\tau(\mathbf{x}_n, \mathbf{x}_o) = \exp(-1/\tau D_{\mathrm{KL}}(q^{(r-1)}(\boldsymbol{\theta}|\mathbf{x}_n)||q^{(r-1)}(\boldsymbol{\theta}|\mathbf{x}_o)))$ – in this case, two data would be regarded as similar if the current estimation of the density network assigns similar posterior distributions to them, which is a natural measure of similarity in this context.

[3]An alternative approach would be to first learn $p(\boldsymbol{\theta}|b(\mathbf{s}) = 0)$ by applying SNPE to a single feature, $f_1(\mathbf{s}) = b(\mathbf{s})$, and to subsequently run SNPE on the full feature-set, but using $p(\boldsymbol{\theta}|b(s) = 0)$ as prior – however, this would 'waste' simulations for learning $p(\boldsymbol{\theta}|b(\mathbf{s}) = 1)$.

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
