[Supplementary Material]

# Appendix

## A Convergence of the log-loss function

Let $K(\mathbf{x}, \mathbf{x}_o)$ be a kernel. We assume that $K \geq 0$ and that $K(\mathbf{x}_o, \mathbf{x}_o) > 0$. Starting from the marginal

$$p(\mathbf{x}) = \int p(\boldsymbol{\theta})p(\mathbf{x}|\boldsymbol{\theta})d\boldsymbol{\theta}$$

we define a weighted version as follows:

$$p_K(\mathbf{x}) := \frac{p(\mathbf{x})K(\mathbf{x}, \mathbf{x}_o)}{\int p(\mathbf{x}')K(\mathbf{x}', \mathbf{x}_o)d\mathbf{x}'} = \frac{1}{Z_K}p(\mathbf{x})K(\mathbf{x}, \mathbf{x}_o),$$

where we assume that the denominator is nonzero and finite. By the law of large numbers:

$$-\frac{1}{N}\sum_n \frac{p(\boldsymbol{\theta}_n)}{\tilde{p}(\boldsymbol{\theta}_n)}K(\mathbf{x}, \mathbf{x}_o)\log q_\phi(\boldsymbol{\theta}_n|\mathbf{x}_n) \xrightarrow{a.s.} \langle -\frac{p(\boldsymbol{\theta})}{\tilde{p}(\boldsymbol{\theta})}K(\mathbf{x}, \mathbf{x}_o)\log q_\phi(\boldsymbol{\theta}|\mathbf{x})\rangle_{\tilde{p}(\boldsymbol{\theta})p(\mathbf{x}|\boldsymbol{\theta})}$$

$$= \langle -K(\mathbf{x}, \mathbf{x}_o)\log q_\phi(\boldsymbol{\theta}|\mathbf{x})\rangle_{p(\boldsymbol{\theta})p(\mathbf{x}|\boldsymbol{\theta})}$$

$$= \langle -K(\mathbf{x}, \mathbf{x}_o)\log q_\phi(\boldsymbol{\theta}|\mathbf{x})\rangle_{p(\mathbf{x})p(\boldsymbol{\theta}|\mathbf{x})}$$

$$= Z_K\langle -\log q_\phi(\boldsymbol{\theta}|\mathbf{x})\rangle_{p_K(\mathbf{x})p(\boldsymbol{\theta}|\mathbf{x})}$$

This expression equals

$$Z_K D_{\text{KL}}\left(p_K(\mathbf{x})p(\boldsymbol{\theta}|\mathbf{x})||p_K(\mathbf{x})q_\phi(\boldsymbol{\theta}|\mathbf{x})\right) + const.,$$

where the constant does not depend on $\phi$. Assuming that the family $q_\phi$ is sufficiently flexible to model the posterior distribution $p(\boldsymbol{\theta}|\mathbf{x})$, the above quantity is minimised iff the two distributions agree, ie. iff

$$p(\boldsymbol{\theta}|\mathbf{x}) = q_\phi(\boldsymbol{\theta}|\mathbf{x})$$

almost everywhere, where $p(\mathbf{x})K(\mathbf{x}, \mathbf{x}_o) \neq 0$.

## B Details on algorithm and optimisation

---

**Algorithm 1: Training SNPE**

---

initialise Bayesian MDN with parameters $\phi = \{\phi_m, \phi_s\}$ and K components
initialise proposal prior $\tilde{p}(\boldsymbol{\theta})^{(1)}$ with prior $p(\boldsymbol{\theta})$
initialise prior over network weights $\pi^{(0)}$ as $\mathcal{N}(\mathbf{w}|\mathbf{0}, \lambda^{-1}\mathbf{I})$

**repeat**

> **for** $n = 1 \ldots N$ **do**
>> sample $\boldsymbol{\theta}_n \sim \tilde{p}(\boldsymbol{\theta})^{(r)}$
>> sample $\mathbf{x}_n \sim p(\mathbf{x}|\boldsymbol{\theta}_n)$
>
> optional: add components to neural network
>
> (re)train Bayesian MDN using Eq.2
>
> set $\hat{p}(\boldsymbol{\theta}|\mathbf{x} = \mathbf{x}_o)^{(r)} := q_\mathbf{w}(\boldsymbol{\theta}|\mathbf{x}_o)$ where $\mathbf{w} = \phi_m$
>
> $\tilde{p}(\boldsymbol{\theta})^{(r+1)} \leftarrow \hat{p}(\boldsymbol{\theta}|\mathbf{x} = \mathbf{x}_o)^{(r)}$

**until** $\hat{p}(\boldsymbol{\theta}|\mathbf{x} = \mathbf{x}_o)$ has converged

---

The precision of the prior on $\mathbf{w}$ is fixed to $\lambda = 0.01$. We normalise importance weights per round. For optimisation, we use Adam with proposed default settings [1]. We rescale gradients such that their combined norm does not exceed a threshold of 0.1 [2].

[1] D Kingma and Ba J. *Adam: A Method for Stochastic Optimization*. In *ICLR*, 2015.

[2] I Sutskever, O Vinyals, and Q V Le. *Sequence to sequence learning with neural networks*. In *Adv in Neur In*, 2014.

## C  Continual learning through the $D_{\text{KL}}$-term

The $D_{\text{KL}}$-term in Eq. 2 implements continual learning. In Bayesian inference calculating the posterior given $r$ rounds is equivalent to taking the posterior after $r - 1$ rounds as the prior for round $r$. Translating this to variational inference formulation gives Eq. 2.

The $D_{\text{KL}}$-term is between two Gaussian distributions over weights $\mathbf{w}$:

$$D_{\text{KL}}(\tilde{q}_\phi^{(r)}(\mathbf{w})||\tilde{q}_\phi^{(r-1)}(\mathbf{w})) = \frac{1}{2}\left[\log\frac{|\Sigma_1|}{|\Sigma_2|} + \text{tr}(\Sigma_1^{-1}\Sigma_2) - d + (\mu_1 - \mu_2)^T\Sigma_1^{-1}(\mu_1 - \mu_2)\right],$$

where $d$ is the dimension of the space (the number of weights) and the parameters of $\tilde{q}^{(r-1)}$ are $\Sigma_1$ and $\mu_1$.

A central term in the $D_{\text{KL}}$ above is a quadratic penalty on the change in posterior mean (of MDN weights) which is weighted by the posterior precision of round $r - 1$. Thus, given that the posterior precision $\Sigma_1^{-1}$ increases with $r$, the penalty on the change in means also increases with rounds.

## D  Details of simulated and neurophysiological data

### D.1  Mixture-models

**Models**  SNPE is applied to two distinct mixture-models. The first is a mixture of two Gaussians with a common mean:

$$p(x|\theta) = \alpha\mathcal{N}(x|\theta, \sigma_1^2) + (1 - \alpha)\mathcal{N}(x|\theta, \sigma_2^2)$$

The second model is a mixture of two Gaussians, such that

$$p(x|\theta) = \alpha\mathcal{N}(x|\theta, \sigma_1^2) + (1 - \alpha)\mathcal{N}(x| - \theta, \sigma_1^2).$$

**Inference**  We set $\alpha = 0.5, \sigma_1 = 1, \sigma_2 = 0.1$. The prior is chosen as $p(\theta) \sim \mathcal{U}(-10, 10)$.

For the first model, we run 6 rounds of SNPE and CDE-LFI with 1000 samples per round. We initialise our SNPE with 2 components.

For the second model we draw 250 samples per round, use 3 rounds, and add a second component to SNPE after the second round.

### D.2  Generalised linear model

**Model**  We simulate the activity of a neuron depending on a single set of covariates. Neural activity is subdivided in bins and, within each bin $i$, spikes are generated according to a Bernoulli observation model:

$$y_i \sim \text{Bern}(\eta(\mathbf{v}_i^\top\boldsymbol{\beta})),$$

where $\mathbf{v}_i^\top\boldsymbol{\beta}$ is the convolution of the white-noise input (represented by $\mathbf{v}_i$) and a linear filter with coefficients $\boldsymbol{\beta}$, and $\eta(\cdot) = \exp(\cdot)/(1 + \exp(\cdot))$ is the canonical link function for a Bernoulli GLM.

**Inference**  We apply SNPE to a GLM with a 10-dimensional parameter vector $\boldsymbol{\beta}$. As summary statistics, we use the cross-correlation between input and response, i.e., the sufficient statistics for the generative model.

A Gaussian prior with mean $\mathbf{0}$ and covariance $\boldsymbol{\Sigma}_{\boldsymbol{\beta}} = \sigma^2(\mathbf{F}^\top\mathbf{F})^{-1}$ is used, where $\mathbf{F}$ encourages smoothness, by penalizing the second-order differences in the vector of parameters [51]. SNPE is run for 5 rounds with 5000 GLM simulations each. We only enforce continuity in MDN weights after round 3, when the proposal distribution converged.

To perform PG-MCMC, the generative model is augmented with latent Pólya-Gamma distributed random variables $\boldsymbol{\omega}$, and samples from $\boldsymbol{\omega}$ and $\boldsymbol{\beta}$ are drawn according to the iterative scheme described by Polson and Scott [42]. The set of samples for $\boldsymbol{\beta}$ represents a draw from the posterior distribution of $\boldsymbol{\beta}$ given the data. PG-MCMC procedure uses the same prior as in the SNPE to estimate the posterior.

### D.3  Single-compartment Hodgkin-Huxley neuron

**Model**  The single-compartment Hodgkin-Huxley neuron uses channel kinetics as in [45]:

$$C_m\frac{dV}{dt} = g_{\text{leak}}(E_{\text{leak}} - V) + \bar{g}_{\text{Na}}m^3h(E_{\text{Na}} - V) + \bar{g}_{\text{K}}n^4(E_{\text{K}} - V) + \bar{g}_{\text{M}}p(E_{\text{K}} - V) + I_{\text{inj}} + \sigma(t),$$

where $C_m$ is membrane capacitance, $V$ membrane potential, $\bar{g}_c$ density of channels of type $c$ ($m$, $h$, $n$, $p$) of the channel gating kinetic variables, $E_c$ reversal potential of $c$, and $\sigma(t)$ intrinsic neural noise. The right hand side is composed of a leak current, a Na-current, a K-current, a slow voltage-dependent K-current responsible for spike-frequency adaptation, and an injected current $I_{inj}$. Channel gating variables have dynamics fully characterised by the neuron membrane potential, once the kinetic parameters are known.

**Inference** We illustrate SNPE in a model with 12 parameters ($g_{leak}, \bar{g}_{Na}, \bar{g}_K, \bar{g}_M, E_{leak}, E_{Na}, E_K, V_T, \sigma, k_{\beta n1}, k_{\beta n2}, \tau_{max}$) (where $k_{\beta n1}$ and $k_{\beta n2}$ control the kinetics of K channel activation, and $\sigma$ is the magnitude of the injected Gaussian noise), and 20 voltage features (number of spikes, resting potential, 10 lagged auto-correlations, and the first 8 voltage moments). SNPE is applied to the log absolute value of the parameters ($\log |g_{leak}|, \log |\bar{g}_{Na}|...$).

The prior distribution over the parameters is uniform, and centred around the true parameter values:

$$\theta \sim \mathcal{U}\left(\frac{1}{2}\boldsymbol{\theta}^*, \frac{3}{2}\boldsymbol{\theta}^*\right)$$

SNPE is run for 5 rounds with 5000 Hodgkin-Huxley simulations each, and we fix the posterior to be a mixture of two Gaussians.

### D.4 Inference in in-vitro recordings from Allen Cell-type database

We apply the approach to *in vitro* recordings from mouse visual cortex (Allen Cell Type Database), (illustration on cell *464212183* in Fig. 3E-G), and inferred the posterior over 12 parameters, as for the application to the simulated data from the Hodgkin-Huxley neuron. We choose the same prior as in the Hodgkin-Huxley simulated data. SNPE is run for 5 rounds with 5000 Hodgkin-Huxley simulations each, and we fix the posterior to be a mixture of two Gaussians.

### D.5 Autapse

**Model** The autapse model corresponds to a neuron synapsing onto itself with connection strength $J$, time constant $\tau$, injected current $I_{inj}$ and external white noise source $\eta_t \sim \mathcal{N}(0, 1)$ with $\langle \eta_s, \eta_t \rangle = \delta_{t,s}$:

$$\tau\frac{dr}{dt} = -r + Jr + I_{inj} + \sigma\eta_t,$$

Using this formula it can be straightforwardly shown that the system has unstable dynamics if $J > 1$.

**Inference** We apply SNPE to infer two parameters of the autapse model $(J, \tau)$, where the feature of interest is the mean across time of the trace. The prior distribution over the parameters is uniform:

$$J \sim \mathcal{U}(0, 2)$$
$$\tau \sim \mathcal{U}(-1, 2.5),$$

where the true parameters are $(0.75, 1)$. We note that the prior allows for the time constant $\tau$ to take negative values: while negative time constants do not make physical sense, we note that these are mathematically equivalent to positive time constants where the autapse equation has flipped signs. We draw 1000 samples for each one of 5 rounds.

# E  Bad features

When sampling from the prior, many parameters sets can lead the Hodgkin-Huxley model to non-spiking behaviour, and therefore features that depend on the presence of spiking, such as latency to first spike, are not defined (Fig. E.1A). In the absence of spiking, the algorithm imputes values to the undefined features, values which are learned during the MDN training. In Fig. E.1B, the imputed value for the latency is close to the mean latency, although we have observed this not to be generally the case.

Figure E.1: **We can impute values to missing features. A.** Hodgkin-Huxley simulations with parameters sampled from the prior, with several parameters sets leading to non-spiking behaviour. **B.** Latency to first spike as a function of firing rate, for samples from the posterior distribution. In this case, the imputed value for the latency (in orange) is close to the mean latency (in green).

# F  Comparison with Sequential Monte-Carlo ABC

SNPE has been tested on problems with 10 or more parameters, whereas most ABC methods (such as SMC-ABC [25]) have addressed problems with fewer parameters, since sampling-based methods require large numbers of simulations. In a GLM with 10 parameters, we observe that our method consistently performs better than SMC-ABC, even when SMC is given orders of magnitude more simulations (Fig. F.1).

Figure F.1: **Comparison between SNPE and SMC-ABC in a GLM with 10 parameters.** The reference (PG-MCMC) posterior means and variances **A.** and covariance **B.** are recovered well by SNPE after 25000 simulations, whereas sequential Monte-Carlo ABC performs worse with over $4 \times 10^6$ simulations. For the application of the SMC-ABC algorithm, we used 1000 particles and a sequence of tolerances $\{\epsilon_i\}_{i=0}^n, \epsilon_i = 15 \times 0.9^i$.

# G Supplementary Figures

Figure G.1: Full posterior inferred for GLM by SNPE. In red, ground-truth parameter values. 2-D marginals for SNPE (lines) and PG-MCMC (histograms). White and yellow contour lines correspond respectively to $68\%$ and $95\%$ of the mass for SNPE.

Figure G.2: Full posterior inferred for HH synthetic data

Figure G.3: Full posterior inferred for HH real data