[Reviews · NeurIPS 2017]

Reviewer 1



The authors present an Approximate Bayesian Computation approach to fitting dynamical systems models of single neuron dynamics. This is an interesting paper that aims to bring ABC to the neuroscience community. The authors introduce some tricks to handle the specifics of neuron models (handling bad simulations and bad features), and since fitting nonlinear single neuron models is typically quite painful, this could be a valuable tool. Overall, this paper is a real gem. My biggest issue is that, in its current form, many of the proposed improvements over [19] are a bit unclear in practice. Sections 2.2-2.3 were difficult to parse for someone less familiar with ABC/MDNs. Since these are the main improvements, it would help to add/clarify here. Additionally, the classifier for bad simulations, the imputation for bad features, and the recurrent network for learning features need to be included in the algorithm in the appendix. Additionally, the value of imputation for missing features in neural models is plausible, but completely theoretical in the paper’s current form. For the HH model, for instance, latency is not a feature and it seems there aren’t good simulations with bad features. Is there a clear example where imputation would be necessary for inference? Minor issues: In section 2. “could” should be replaced with “here we do.” One challenge in reading the paper is that many of the improvements seem to be purely hypothetical, until you get to the results. It would be helpful to have a more complete description of the features used in the main text, and also to clarify at the beginning of section 2.1 that x=f(s). Using a recurrent neural network for learning features seems like an interesting direction, but the results here are more of a teaser than anything else. Completely unpacking this could likely be a paper by itself. For the GLM simulation -- what features are used? Also, the data augmentation scheme could be more clear in the appendix. If I understand correctly it isn’t necessary for the generative model, but just for getting the true posteriors and in PG-MCMC.

Reviewer 2



The manuscript describes a method to learn models for single neuron firing rates when the resulting loss function is some aggregation function such as the firing rate or the latency. They employ approximate Bayesian computation to solve the problem, when there are multiple such loss functions to consider. This seems a niche problem that may not be of great interest to the community. Furthermore, the problem becomes less important with the advent of back-propagation tools that allow optimization of practically any aggregation function.

Reviewer 3



The authors demonstrate that by making practical extensions to the Approximate Bayesian Computation (ABC) framework that they can develop computationally-feasible posterior estimates on mechanistic neural models. Given the dramatic increase in cellular and neural data that is supposed to occur over the next several years, this type of approaches could be useful for understanding the diversity of neurons. Overall, the paper is mostly clear and has reasonable contributions that should be valued at this conference. However, I do have some concerns about the paper and its details. First, in Equation 2 the authors propose a framework to smooth and incorporate historical information from previous simulation rounds by using a KL-divergence term to encourage posterior simulations to be similar between rounds. I do not believe that this method would converge; typically, this type of approach has an increasing multiplier on the KL-divergence term that increases by round to stabilize the learning and give convergence properties. The authors should discuss this a bit more, and elaborate on when this approach will actually work. Second, I do not understand the how the RNN in the propose feature learning works. Clearly it would be useful to let an ML approach learn appropriate features instead of requiring hand-selected features. However, I don't see what the RNN is trained against, or how the RNN is encouraged to learn good features. I would encourage the authors to expand this section and clarify its contributions.

Reviewer 4



The authors significantly extend the likelihood-free inference approach of [19] in many ways to make it more powerful. It consists of incorporating several clever tricks. The inferred posterior results are impressive, but I wonder how well the algorithm scales to large datasets and higher-dimensional parameter spaces. Future studies in establishing relevant theories would be very useful. 1. 'can not' vs 'cannot' In English, the two are not the same: - 'can not': it is possible to not - 'cannot': it is not possible to I think the authors meant 'cannot' when they wrote 'can not' in the text. 2. 'neural dynamics' in the title The authors only demonstrated in a relatively low-dimensional system with small datasets, i.e., single neurons. Though they claimed it scales to large dimensions, they have not demonstrated this; in particular, they have shown anything about population dynamics. I think the title is overselling by saying 'neural dynamics' and suggest changing it to single neurons (or properly show that it scales up). 3. missing details Although some of the derivations are presented in the appendix, the details regarding the parameterization of the mixture-density network (MDN) and the automatic feature learning network (which just says GRU's in the figure).